# Epitranscriptomics: An Additional Regulatory Layer in Plants’ Development and Stress Response

**DOI:** 10.3390/plants11081033

**Published:** 2022-04-11

**Authors:** Yasira Shoaib, Babar Usman, Hunseung Kang, Ki-Hong Jung

**Affiliations:** 1Graduate School of Biotechnology and Crop Biotech Institute, Kyung Hee University, Yongin-si 17104, Korea; yasirashoaib@gmail.com (Y.S.); babarusman119@gmail.com (B.U.); 2Department of Applied Biology, College of Agriculture and Life Sciences, Chonnam National University, Gwangju 61186, Korea; hskang@jnu.ac.kr

**Keywords:** epitranscriptomics, m^6^A, plant development, biotic and abiotic stress

## Abstract

Epitranscriptomics has added a new layer of regulatory machinery to eukaryotes, and the advancement of sequencing technology has revealed more than 170 post-transcriptional modifications in various types of RNAs, including messenger RNA (mRNA), transfer RNA (tRNA), ribosomal RNA (rRNA), and long non-coding RNA (lncRNA). Among these, N6-methyladenosine (m^6^A) and N5-methylcytidine (m^5^C) are the most prevalent internal mRNA modifications. These regulate various aspects of RNA metabolism, mainly mRNA degradation and translation. Recent advances have shown that regulation of RNA fate mediated by these epitranscriptomic marks has pervasive effects on a plant’s development and responses to various biotic and abiotic stresses. Recently, it was demonstrated that the removal of human-FTO-mediated m^6^A from transcripts in transgenic rice and potatoes caused a dramatic increase in their yield, and that the m^6^A reader protein mediates stress responses in wheat and apple, indicating that regulation of m^6^A levels could be an efficient strategy for crop improvement. However, changing the overall m^6^A levels might have unpredictable effects; therefore, the identification of precise m^6^A levels at a single-base resolution is essential. In this review, we emphasize the roles of epitranscriptomic modifications in modulating molecular, physiological, and stress responses in plants, and provide an outlook on epitranscriptome engineering as a promising tool to ensure food security by editing specific m^6^A and m^5^C sites through robust genome-editing technology.

## 1. Introduction

Recently, biochemical modifications of RNAs designated as epitranscriptomes have added a new layer of regulatory networks to DNA methylation and histone modifications by fine-tuning key developmental processes and stress responses in all living organisms [1,2,3]. In eukaryotes, covalent RNA modifications add ubiquitous layers of information, and more than 170 post-transcriptional RNA modifications are found in different cellular RNAs, including messenger RNA (mRNA), ribosomal RNA (rRNA), transfer RNA (tRNA), long non-coding (lncRNA), micro RNA (miRNA), and small interfering RNA (siRNA) [4,5]. RNA modifications were studied exclusively in non-coding RNAs, including rRNA, tRNA, and small nuclear (snRNA), and they play critical roles in stabilizing structural conformations and modulating base pairing [6,7]. However, the advancement of genomic approaches has led to the discovery and mapping of various mRNA modifications and the elucidation of their roles in regulating the mRNA fate [8,9]. Many of the RNA modifications and enzymes responsible for installation and removal are highly conserved across all three domains of life [10], suggesting the importance of complex and flexible RNA regulation. The loss of these modifications affects key regulatory processes and can cause multiple human diseases [11,12] as well as defective development and stress responses in plants [13,14]. The RNA epitranscriptomic modification profiles vary according to the organs, cell types, and developmental stages under study. In *Arabidopsis thaliana*, transcriptome-wide N6-methyladenosine (m^6^A) sequencing revealed that 33.5% of transcripts exhibited differential m^6^A levels in leaves, roots, and flowers [15]. Consistently, the transcript levels of m^6^A writers, erasers, and reader proteins vary in different tissues and at different developmental stages [16]. Therefore, the effects of m^6^A and other epitranscriptome regulatory marks on plant development and abiotic and biotic stress responses will be an important avenue of future inquiry. The diverse eukaryotic modifications found on mRNA, tRNA, and rRNA are at first briefly described here.

### 1.1. mRNA Modification

The various modifications found in mRNA include m^6^A [17], 7-methylguanosine (m^7^G) [18], 2′-O, *N*6-dimethyladenosine (m^6^Am) [19], *N*1-methyladenosine (m^1^A) [20], 4-acetylcytidine (ac4C) [21], 5-hydroxymethylcytidine (5hmC) [22], 2′-O methylation at any nucleotide (Nm) [23], Inosine (I) [24], and pseudouridine (Ψ) [5]. They do not alter the coding capacity of mRNA [25], but they do alter the chemical properties of the transcripts, thereby affecting base pairing and the formation of ribonucleoprotein complexes [1]. Among these, m^6^A is the most abundant and dynamic mRNA internal modification in more than 5000 transcripts and is installed, erased, and read by methyltransferases, demethylases, and RNA binding proteins, respectively [5,24,25]. It affects transcripts’ fate and translation [26,27,28] in the nucleus and cytoplasm by regulating a wide variety of cellular processes including RNA turnover [29,30,31,32], stability [33,34], mRNA abundance [35], 3′UTR processing [36], alternative polyadenylation [37,38], mRNA splicing [39], and primary micro RNA processing [40]. In chloroplast and mitochondria, about 98–100% and 86–90% of transcripts are m^6^A-modified, respectively, with approximately 4–6 m^6^A sites per transcript [41]. RNA metabolism, including the processing, splicing, editing, and decaying processes, is important for the genetic regulation of chloroplasts and mitochondria, which is essential for plants’ survival and responses to changing environmental conditions [42,43]. RNA analysis of terrestrial plants has shown that RNA editing involving cytosine-to-uracil (C–U) transitions occurs prevalently in mitochondria and chloroplasts and affects their coding sequences, RNA structure, splicing and stability [44]. Transcriptome-wide profiling by m^6^A RNA sequencing has revealed the abundance of m^6^A near the stop codon and 3′UTR in the RRACH consensus sequence, which is highly conserved in various organisms [8,45,46] and in the UGUAY motif found exclusively in plants [16,33]. Another important epitranscriptomic mark located mainly at the 3′UTR and mRNA coding regions is 5-methylcytosine (m^5^C), which plays an essential role in development and stress responses, but very limited information is available on its roles in plants [47,48].

### 1.2. tRNA Modification

The tRNA is the richest source of modifications, and on average, 25% of its nucleotides are modified, presenting the greatest chemical variety and complexity [49,50]. These modifications range from simple methylation to complex multi-step transformations and include the incorporation of a range of low-molecular-weight metabolites [50]. The various tRNA modifications include ribose 2-O methylation, pseudouridine (ψ), dihydrouridine, methylguanosine (m^1^G, m^2^G, and m^7^G) [51], dimethylguanosine (m^2,2^G) [52], N1-methyladenosine (m^1^A) [53], 3-amino-3-propylcarboxyuridine (acp3U) [54], cytosine 2′-O-methylation (C_m_) [55], and RNA editing from adenosine to inosine (A–I). m^5^C and 1-methylguanidine (m^1^G) are the most abundant tRNA modifications [2,56]. The post-transcriptional modifications of tRNA are critical for protein translation and proper cell growth [52,57]. The thirty-fourth and thirty-seventh wobble positions of the tRNA anticodon loop are most frequently modified, and they play essential roles in tRNA’s structure and function, codon recognition, and decoding, along with its translation initiation and elongation processes [58]. Deficiencies in nucleoside modification produce a wide variety of effects ranging from decreased virulence in bacteria, disease of the neural system in humans, and regulation of gene expression and stress responses in plants. A recent study revealed that more than 90 tRNA-modifying enzymes are present in the Arabidopsis genome. Modification genes of tRNA including *AtTRM10*, *AtTRM11*, *AtTRM82*, *AtKTI12,* and *AtELP1*, are responsible for modifications including m^1^G, m^2^G, m^7^G, and ncm^5^U, respectively [49]. In Arabidopsis and rice, tRNA marks increase in response to abiotic stresses including drought, salt, or cold temperatures [57,59].

### 1.3. rRNA Modification

The rRNA methylation processes evolved to refine the rRNA structure and optimize its functions [60,61]. In Arabidopsis, the various rRNA modifications include ribose 2′-O-methylation, pseudouridine (ψ), and base modifications including m^1^N, m^6^N, m^7^N, m^3^U, and acp3N [51,62]. Most of the modified bases in rRNA are located at the interface between the large and small subunits of the ribosome, corresponding to the P-site and the A-site [63]. Pseudouridine (ψ) and 2′-O-ribose methylation are regulated by small nucleolar ribonucleoprotein (snoRNPs) complexes that direct sequence-specific targeting. In contrast, rRNA base modifications are catalyzed by site-specific base methyltransferases [51]. The ribose 2′-O methylation can stabilize rRNA–mRNA, rRNA–tRNA, and rRNA–protein interactions [62]. Methylation in the rRNA of the chloroplast plays a vital role in regulating translation, and it has recently been established that chloroplast MraW-like (CMAL) protein catalyzes the m^4^C methylation of C_1352_ in the chloroplast’s 16S ribosomal subunit and plays an important role in ribosome biogenesis, plant development, and hormonal responses in Arabidopsis. RsmD is a chloroplast-localized m^2^G methyltransferase that affects plant developmental processes under cold stress [64].

This review focused mainly on the roles of m^6^A and m^5^C in the context of regulating molecular, cellular, and physiological processes during plants’ development and responses to stress. The robust genome editing technology for improving crops quality, survival, and productivity has highlighted future implications of m^6^A and m^5^C editing.

## 2. Developmental and Physiological Regulation by Epitranscriptomic Modifications in Plants

Epitranscriptomic modifications play important roles during plant development and in various responses to biotic and abiotic stresses. The major developmental processes affected by these modifications include organogenesis, embryonic and cotyledon development, seed development and seed yield, root and shoot growth, leaf morphology, trichome branching, floral transition, the proliferation of shoot apical meristem, and fruit ripening, as illustrated in Table 1.

### 2.1. Seed Development

Seed development is a complex process integrating different genetic, metabolic, and physiological pathways regulated by transcriptional, epigenetic, peptide hormone, and sugar regulators [75,76]. The chemical modifications associated with seed development, such as oxidation and methylation in mRNA and genomic DNA, affect gene expression during the later stages of seed development. DNA methylation in Arabidopsis is a dynamic process, and during seed development, there is a drastic increase in the global level of non-CG methylation throughout the seed, whereas CG and CHG-methylations do not change significantly. DNA methylation regulates the maternal expression of *DOG4* and *ALN*, which are the negative regulators of seed dormancy. However, the special methylation marks associated with seed dormancy and the germination transcriptomes remain to be elucidated [77]. MTA, an m^6^A mRNA methyltransferase, is essential for embryogenesis, and its homozygous insertional knockout mutant “mta” showed an embryo arrest at the globular stage due to a lack of m^6^A at the poly(A) RNA, whereas the hemizygotes produced green and white seeds in immature siliques. However, the complementation lines rescued the embryo-lethal phenotype, indicating that the insertion mutation in *MTA* was embryo-lethal [69,78]. *AtTRM61* and *AtTRM6* cause N1 methylation of adensoine58 (A58) in tRNA, and the loss of function of either of these tRNA methyltransferases causes seed abortion. Mutations in the complex *AtTRM61*/*AtTRM6* subunits result in developmental defects in the embryo and endosperm. However, conditional complementation of At*TRM61* showed that tRNA m1A58 modification is crucial for endosperm and embryo development [79]. *CMAL* is responsible for the methylation of N4-methylcytidine rRNA in the chloroplast and plays a key role in the chloroplast’s function, development, and abscisic acid (ABA) response in Arabidopsis. The loss-of-function *cmal* mutant exhibited a reduction in silique size, the number of seeds per silique, and total seed yield compared with wild-type (WT) plants, indicating its important role in seed development [80].

### 2.2. Root Development and Growth

Root development is a critical aspect of plant growth and allows the effective use of water resources. Plants, being sessile by nature, must adapt to various environmental cues. Epitranscriptomic modifications play a crucial role in root development processes. In Arabidopsis, *AtTRM4B* is involved in the methylation of m^5^C sites in the root transcriptome and positively regulates its growth through cell proliferation of root apical meristem. A T-DNA insertion mutant, *trm4b*, had a shorter primary root than the WT. The *trm4b*/*trdmt1* double mutant also exhibited a shorter root phenotype. Furthermore, the *TRM4B* mutant was more sensitive to oxidation stress, implying that *TRM4B* contributes to root growth by regulating the response to oxidative stress [47]. Another study has shown that *TRM4B* contributes to primary and lateral root development in Arabidopsis by regulating the transcript levels of *SHY2* and *IAA16*. The m^5^C levels in *TRM4B* were reduced by 20–30% in roots and exhibited a shorter root phenotype; however, its level remains unchanged in aerial tissues [81]. At*TRM5* is a bifunctional guanine and inosine-N1-methyltransferase tRNA and *trm5-1* mutant with reduced levels of m^1^G and m^1^I and a reduced number of lateral roots and total root length compared with WT plants. However, *TRM5* complementation lines reversed the knockout mutant phenotypes, indicating that *TRM5* is involved in regulating the root development of Arabidopsis [67]. The m^6^A writer and reader proteins are highly expressed in the root meristems, apexes, and lateral root primordia [69,73,82]. In poplar, root development is affected by *PtrMTA* and *OE-PtrMTA-14*, *OE-PtrMTA-10*, and *OE-PtrMTA*-6 lines with almost double the m^6^A level, exhibiting better root and root tip growth compared with those of WT [83]. Recent research showed that the m^6^A level changes in response to ammonium (NH4^+^) nutrition and regulates the proteome response through altered translation in maritime pine roots [84]. Rice cultivar (cv.9311) exposed to cadmium stress exhibited abnormal root development caused by altered methylation profiles in transcripts involved in various biosynthetic, metabolic, and signaling processes, indicating that m^6^A plays an important role in regulating the gene expression level of various cellular pathways [85]. In Arabidopsis, correct m^6^A methylation plays an important role in developmental decisions, and *Virilizer-1* (m^6^A methyltransferase) plays an important role in maintaining m^6^A levels. The deletion of *vir-1* showed aberrant root cap formation and defective protoxylem development, indicating that m^6^A is essential for root development [73]. Another study has shown that multi-walled carbon nanotubes inhibit root growth by reducing m^6^A levels [86]. In rice, FTO expression increases root apical meristem cell proliferation and modulation of m^6^A RNA levels, which is a promising strategy to improve growth. *FTO*-transgenic plants showed a 35% and 45% increase in the total number and length of their lateral roots, respectively, and the number and length of their primary roots increased more than 3.3 fold at the tillering stage compared with the WT plants [87]. The m^6^A reader proteins named *ECT2*, *ECT3,* and *ECT4* are highly expressed at the root apex and throughout the lateral root formation. Loss of ECT2 function caused a right-ward tilt in root growth, and the ect2/ect3 double mutants show slower root growth, whereas the ect2/ect3/ect4 triple mutants show agravitropic behavior along with a slower root growth compared with the WT [82]. Genes affecting various plant developmental processes such as floral transition [26,88], seed development [37,74,77], root growth [51,77], leaf growth [82], and fruit ripening [15,78] are illustrated in Figure 1.

### 2.3. Anther/Pollen Development

Anthers produce male gametes and certain sporophytic and gametophytic tissues in flowering plants. The tapetum of anthers acts as a bridge for nutrient exchange and communication between sporophytic and gametophytic cells. A recent study has shown the involvement of m^6^A in anther development in rice. *OsEMD2L* contains an N6-adenine methyltransferase-like (MLT) domain, and the *osemd2l* mutant showed an altered m^6^A landscape with Eternal Tapetum 1 (*EAT1*) transcription. The dysregulated alternative splicing and polyadenylation of *EAT1* resulted in the suppression of *OsAP25* and *OsAP37* and led to delayed tapetal-programmed cell death and male sterility [71]. Another study showed that the transgenic expression of *FTO* in rice increased the total number of productive tillers per plant by 42% and improved productivity [87]. *OsFIP* and *OsMTA2* are the components of the m^6^A RNA methyltransferase complex in rice. *OsFIP* is essential for male rice gametogenesis and modifies m^6^A during sporogenesis by recognizing a panicle-specific “UGWAMH” motif. The *osfip* knockout mutant showed an early degeneration of microspores and abnormal meiosis in prophase I, and had 1.4 tillers per plant compared with 4.7 in WT plants. Furthermore, at the late reproductive stage, *fip* plants were almost sterile and had shorter panicles and reduced seed numbers, and 84.8% of the pollen grains lacked starch, indicating that *OsFIP* plays an important role in microspore development [18]. In tomatoes, the widely spread m^6^A modification in anthers is disrupted under cold stress conditions and affects the expression level of genes involved in tapetum and microspore development. The moderately low-temperature-induced pollen abortion is due to impaired micro gametogenesis, tapetum degeneration, and pollen wall formation. Additionally, m^6^A is associated with ABA transport in anthers or sterol accumulation for pollen wall formation, and targets the ATP-binding cassette G gene, *SLABCG31* [89].

### 2.4. Floral Regulation

Precise initiation of flowering is essential for plant reproductive success, and several epigenetic modifications play important roles during floral transition. A recent report showed that m^6^A-mediated RNA modification was involved in the complex genetic regulation that controls floral regulation. The loss of function of the RNA demethylase, *ALKBH10B,* increased m^6^A modification and delayed floral transition due to the increased mRNA decay of the flowering regulator *FT* and its up-regulators, *SPL3,* and *SPL9* [26]. *AtTRM5* encodes nuclear-localized bifunctional tRNA guanine and inosine-*N1*-methyltransferase and is important for growth and development. The loss-of-function *Attrm5* mutant showed an overall slow growth and delayed flowering. At the inflorescence emergence stage, *trm5-1* plants exhibited a reduced number of rosette leaves, smaller leaves, reduced fresh weight, and took longer to flower; however, *TRM5*-overexpressing plants flowered slightly earlier than WT. The delayed flowering phenotype in *trm5-1* mutants was due to a deficiency in floral time regulators, including *GI*, *CO*, and *FT*, the downstream floral meristem identity gene *LEAFY* (*LFY*), and circadian clock-related genes [67]. *CMAL* is a chloroplast-localized rRNA methyltransferase and is responsible for the modification of N4-methylcytidine (m^4^C) in 16S chloroplast rRNA. The loss-of-function *cmal* mutant showed stunted growth and delayed flowering due to altered expression levels of various flowering-related genes, including *APETALA1 (AP1)*, *SUPPRESSOR OF OVEREXPRESSION OF CONSTANS1 (SOC1)*, *FRUITFULL (FUL)*, *CAULIFLOWER (CAL)*, and *Flowering Locus C (FLC)*; however, the *CMAL* complementation lines recovered stunted growth phenotypes, indicating that stunted growth is due to the lack of m^4^C modification [90].

## 3. Biotic and Abiotic Regulation by Epitranscriptomic Modifications in Plants

RNA methylation plays an important role in the response to various environmental stresses by regulating the expression level of key stress-responsive genes. Drought stress increased the m^6^A marks in the 5′UTR region and promoted the translation of several drought-resistant transcripts. Cellular stress also altered the characteristic distribution of m^6^A and metagenic analysis followed by heat shock, and showed that m^6^A was highly enriched in the 5′UTR region in heat-shocked cells compared with the control [28]. Recently, anther development in tomatoes was modulated by the m^6^A-mediated expression level of several pollen development-related genes under a low temperature [91]. In tea (*Camellia sinensis*), m^6^A regulatory genes play an important role in resistance to environmental stresses and the withering process of tea. Drought stress decreased the expression level of several m^6^A writers, including CsMTB1, CsMTC, CsMTA1, CsMTA2, CsMTB2, and CsVIR2, whereas the level of m^6^A erasers and readers was enhanced. Furthermore, the interaction of the methylation regulatory genes of RNA and DNA methylation formed a negative feedback loop, indirectly inhibited flavonoid biosynthesis, and improved the palatability of the oolong tea during the withering process [92]. *CIMTB* is an m^6^A methyltransferase in watermelon, and it helps in adapting to drought stress by regulating reactive oxygen species (ROS) scavenging, photosystem components, phytohormones, and multiple stress-responsive transcription factors [93]. The global m^6^A levels in Arabidopsis increase in response to salt stress, and its dysregulation disrupts the salt stress tolerance mechanism. The m^6^A writer mutants, including *mta*, *mtb*, *vir*, and *hakai*, exhibited salt-sensitive phenotypes in an m^6^A-dependent manner. *VIR*-mediated m^6^A methylation modulated ROS homeostasis by down-regulating the mRNA stability of key salt stress negative regulators, including *GI*, *ATAF1*, and *GSTU17*, by affecting 3′UTR lengthening [36]. In Arabidopsis *alkbh10b* mutants, global m^6^A levels increased under salt stress conditions and exhibited salt-tolerant phenotypes caused by the decreased expression level for several negative regulators of salt stress, including *ATAF1*, *MYB73*, and *BGLU22* [91]. m^6^A regulates the expression level of some transcription factors, including WRKY81 and heat shock proteins (HSP70) in tomatoes during chilling injury, and helps the plant combat cold stress [94]. In wheat, YTH domain-containing RNA-binding m^6^A reader proteins are regulated by various abiotic stresses [95].

It has recently been established that m^6^A plays an important role in regulating the life cycle of various viruses by modifying viral and host RNAs. However, there is conflicting evidence on the role of m^6^A in regulating the viral life cycle. Recent research showed that m^6^A suppresses the replication of rice black-streaked dwarf viruses and is associated with viral persistence in its insect vector [96]. *ALKBH9B* is an m^6^A RNA demethylase in Arabidopsis and its demethylation activity affects the infectivity of alfalfa mosaic viruses (AMV). The suppression of *atalkbh9b* increased the relative abundance of m^6^A in the AMV genome and impaired virus accumulation and systemic invasion of the plant. Therefore, m^6^A modification may act as a regulatory strategy in plants by controlling cytoplasmic-replicating RNA viruses [97]. Infection with a cucumber green mottle mosaic virus (GGMMV) significantly decreased m^6^A levels of 422 differentially methylated transcripts in watermelon because of the increased expression level of the m^6^A demethylase gene *ClALKBH4B*. The decreased m^6^A levels enhanced the transcription of several defense response factors involved in virus-induced gene silencing, such as transcription factors, carbohydrate allocation, and signaling genes, and ultimately activated the immune responses of the plant in the early stages of GGMMV infection [93]. In apples, the m^6^A reader protein, *MhYTP2*, conferred resistance to powdery mildew by regulating the stability of *MdMLO19* mRNA and the translation efficiency of several antioxidant genes [98]. In rice, m^6^A is involved in the conidiation and virulence of the rice blast fungus *Pyriculariaoryzae*, and the N6-adenosine-methyltransferase (*PoIme4*), m^6^A mRNA demethylase (*PoALKB1*), and m^6^A binding proteins (*PoYth1* and*PoYth2*) are involved in the virulence of rice in *P. Oryzae* [99]. The m^6^A modification levels of rice mRNA increased in genes that were expressed at low levels during the viral infection of plants. This modification regulates the expression level of key antiviral genes involved in RNA silencing, resistance, and fundamental antiviral phytohormone metabolic pathways [100]. However, another study showed that in tobacco, the m^6^A level is associated with the tobacco mosaic virus, and its infection increased the expression level of potential demethylase XM-009801708 on the twelfth and twenty-first days of infection, thereby decreasing m^6^A levels [101]. These conflicting pieces of evidence need to be addressed further to illustrate the m^6^A-regulated mechanism in defensive responses to viral infections. Several plant viruses containing a single-stranded RNA genome contain an ALKB domain in their genome and have evolved mechanisms to respond to the regulation of the host m^6^A system [102]. The regulatory roles of post-transcriptional modifications in various stress responses are illustrated in Table 2.

## 4. Molecular Regulation by Epitranscriptomic Modifications in Plants

m^6^A is important in the regulation of various aspects of mRNA metabolism, and m^6^A readers play a precise and complex regulatory role by recognizing changes in m^6^A modification in mRNA. The fate of modified transcripts depends on the reader protein recognizing the modification that may cause the export of modified transcripts from the nucleus to the cytoplasm, where they may be translated, stored in granules, or decayed by the P-bodies [2], as shown in Figure 2.

A recent study showed that the modification of m^6^A regulates alternative polyadenylation (APA) of nitrate signaling-related genes through recognition by the CPSF30-L reader protein [37]. A recent study provided the evidence for m^6^A-regulated protein translation, illustrating that m^6^A is deposited in the 3′-UTR region in response to ammonium nutrition and is correlated with poly(A) lengthening and transcript abundance, thereby an optimal response to the N-supply in maritime pine roots is acquired [84]. An m^6^A reader protein, ECT2, functions in polyadenylation and 3′UTR processing in the nucleus by selectively binding to the m^6^A-containing poly(A) signal FUEs and recruiting polyadenylation machinery to promote mRNA stability and regulate the morphology of trichomes in Arabidopsis [33]. In *Zea mays*, transcriptome-wide m^6^A–mRNA profiling has shown that m^6^A is widely distributed in thousands of protein-coding genes primarily enriched in the 3′UTR region, and strongly correlated with protein translation, and regulates gene expression [106]. Additionally, it also positively regulates the translation of mitochondrial transcripts in Arabidopsis and Brassica [107]. The YTHDF1 and YTHDF3 reader proteins recognize m^6^A residues in the 3′UTR region and enhance translation through interaction with initiation factors or ribosomal subunit proteins [108,109,110]. However, YTHCF2 proteins decrease the amount of m^6^A-modified mRNA in translatable fractions by causing degradation through the sequestering of modified transcripts in processing bodies [109]. Furthermore, in Arabidopsis, m^6^A enhances the stability of transcripts during the salt stress response through the widespread prevention of ribonucleolytic cleavage and an enhanced abundance of salt and osmotic stress-related transcripts [111]. Recently, m^6^A has been found to regulate miRNA processing through the interaction between m^6^A-methyltransferase and TGH, a miRNA biogenesis factor [27]. Being another important internal modification of mRNA, m^5^C also tends to stabilize and translate modified transcripts. Recent research showed that m^5^C methylation promoted the translation of heat-induced mRNAs and induced heat resistance in WT rice plants at high temperatures [103]. Another study showed that m^5^C-containing mRNAs were more stable and enriched in the fractions of graft-mobile transcripts that move from root to shoot, indicating that m^5^C plays an important role in the mobility of transcripts throughout the plant body [68]. There is still a need to further explore m^5^C-mediated mRNA metabolism, especially for specific transcripts involved in plant development and stress responses. The molecular and physiological processes regulated by post-transcriptional modifications are illustrated in Figure 3; however, multiple RNA regulatory mechanisms mediated by epitranscriptomic modifications still need to be discovered.

## 5. Future Perspectives

Climate change and the increase in world population have caused a great threat to sustainable food production; however, recent discoveries on the involvement of post-transcriptional modifications in the regulation of RNA metabolism, plant development, and stress responses have shown that it has a great potential to improve crop survival and productivity, but it is limited by some factors. Firstly, the modulation of epitranscriptomic machinery may cause changes in global m^6^A levels, causing unpredictable effects. Thus, transcriptome-wide mapping at a single-base resolution is critical for precise m^6^A editing without affecting the overall m^6^A levels or the sequences of genes involved in crop development and stress responses [35]. Various advanced sequencing techniques including miCLIP (m^6^A individual-nucleotide-resolution cross-linking and immunoprecipitation) [112], Mazter-seq (RNA digestion via m^6^A-sensitive RNase and sequencing) [113], Nanopore DRS (Direct RNA Sequencing) [114], and m^6^A REF-Seq (m^6^A-sensitive RNA-endoribonuclease-facilitated sequencing) may facilitate the accurate mapping of m^6^A modifications at the cellular level [115].

The second challenge is the precise addition or removal of m^6^A at a specific site in transcripts, which can be resolved by recent advances in CRISPR technology that have revolutionized the editing capabilities of the entire genome [115]. Recently, a new CRISPR-based m^6^A editing system was proposed, in which m^6^A enzymes (writers/erasers) are fused to a dCas13 protein that binds to the targeted RNA transcripts without mediating their cleavage, and the fused m^6^A writers or erasers can add or remove the m^6^A modification at the target site. Furthermore, editing of the transcript target site requires dynamic m^6^A modification maps at a single-base resolution, which are useful for designing sgRNA [102]. Therefore, editing m^6^A modifications in key transcripts involved in growth, development, photosynthesis, and biotic (viral and bacterial infections) and abiotic stress (salt, heat, drought) responses through CRISPR technology may be a feasible method to improve crops’ productivity and stress resistance [102].

Additionally, a CRISPR-based strategy has been developed that can induce targeted epigenetic modifications in DNA by fusing the epigenetic modifier with dCas9. Targeted epigenetic modifications and related phenotypes are stably transmitted to subsequent generations even without transgenes [116,117]. Gallego-Bartolome et al. developed a CRISPR/dCas9 based targeted demethylation system using TET1cd and a modified SunTag system, in which TET1cd (a DNA demethylase) up-regulates the expression level of the FLOWERING WAGENINGEN (*FWA*) gene. This approach was successful in achieving the targeted removal of 5mC at specific loci with high specificity and minimal off-target effects.

The various processes that can be modified by epitranscriptome engineering are shown in Figure 4.

Another approach toward epitranscriptome engineering is through manipulation of the activities of RNA modification-related enzymes in crops. Recently, it has been reported that over-expression of human RNA demethylase (FTO) in rice has up-regulated several pathways related to photosynthesis and nitrogen regulation and greater crop yields by increasing the plants’ root cell proliferation, tiller numbers, photosynthesis rate, and drought resistance [87]. Furthermore, over-expression of an m^6^A-methyltransferase has increased drought resistance in poplars by increasing trichome branching in roots [83], suggesting potential applications of epitranscriptome manipulation to improve survival and productivity.

mRNA modifications such as m^6^A and m^5^C are highly conserved in many plant species, thus epitranscriptome engineering is a promising tool in crop breeding. However, it is essential to understand the specific and generic functions of mRNA modifications and discover additional epitranscriptome components and their associated regulatory mechanisms. The integration of advanced sequencing techniques involving a single-base resolution into genome editing and genetic transformation holds great promise for epitranscriptome engineering to improve food security despite climate change and the global population increase.

## Figures and Tables

**Figure 1 plants-11-01033-f001:**
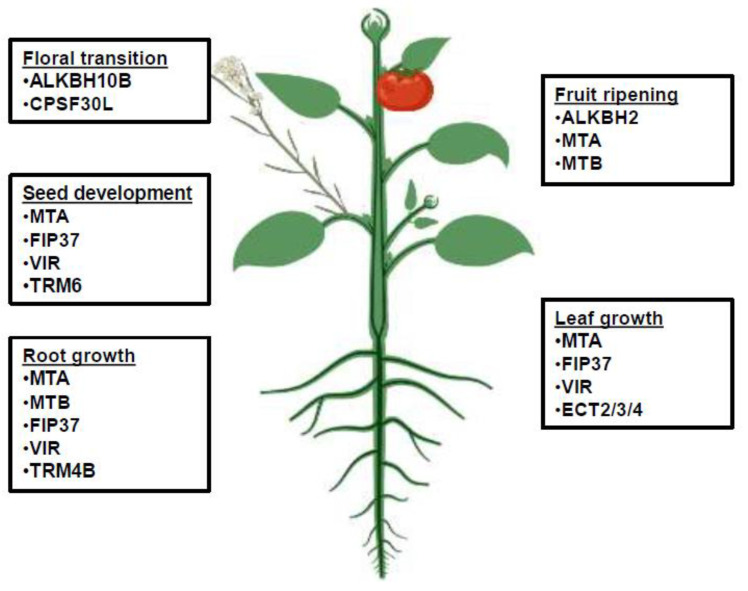
Regulation of plant’s development by post-transcriptional modifications (m^6^A and m^5^C); post-transcriptional modifications regulated by different writer, eraser, and reader proteins affect various plant developmental processes including seed development, leaf and root growth, floral transitions, and fruit ripening.

**Figure 2 plants-11-01033-f002:**
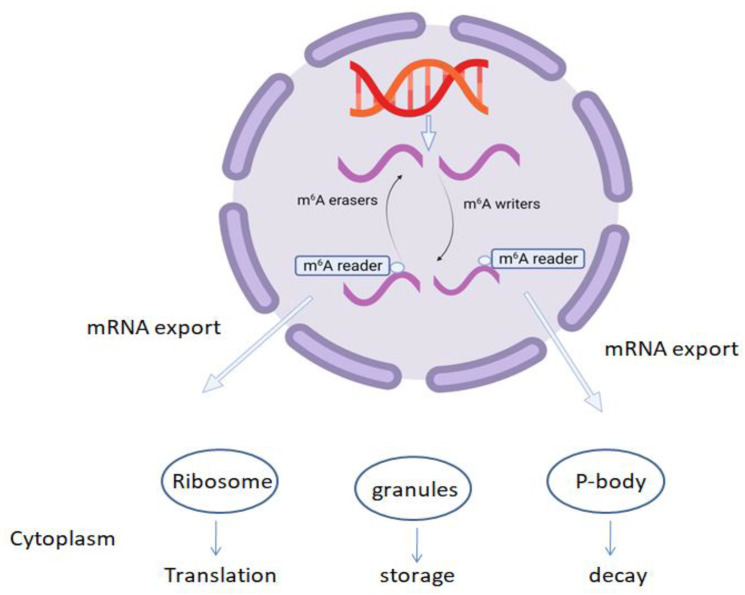
Various cellular processes affected by m^6^A RNA methylation; m^6^A RNA reader proteins determine the fate of modified transcripts and direct their export from the nucleus to the cytoplasm, where transcripts may be translated by the ribosomes, and stored in the granules, or decayed by the P-bodies.

**Figure 3 plants-11-01033-f003:**
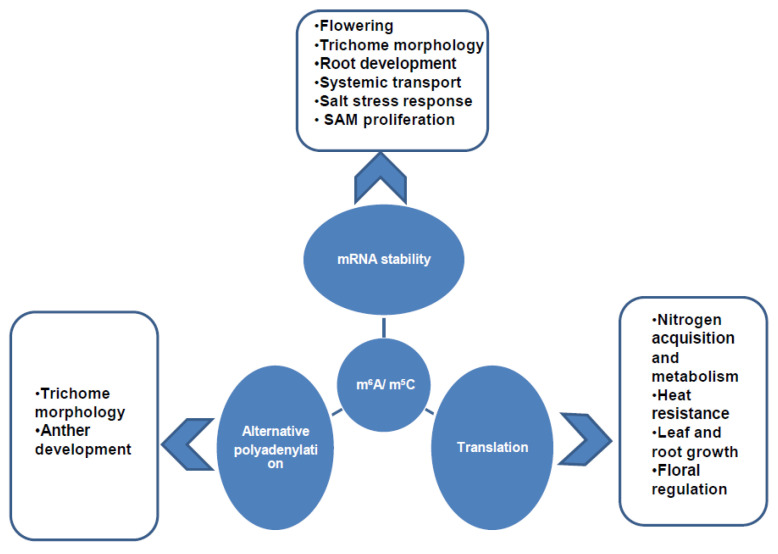
Molecular and physiological regulation by m^6^A and m^5^C; m^6^A and m^5^C modifications regulate various molecular processes including mRNA stability, translation and alternative polyadenylation that affect plant growth and development processes such as anther development, trichome morphology, flowering, SAM proliferation, leaf growth, root growth, and stress responses.

**Figure 4 plants-11-01033-f004:**
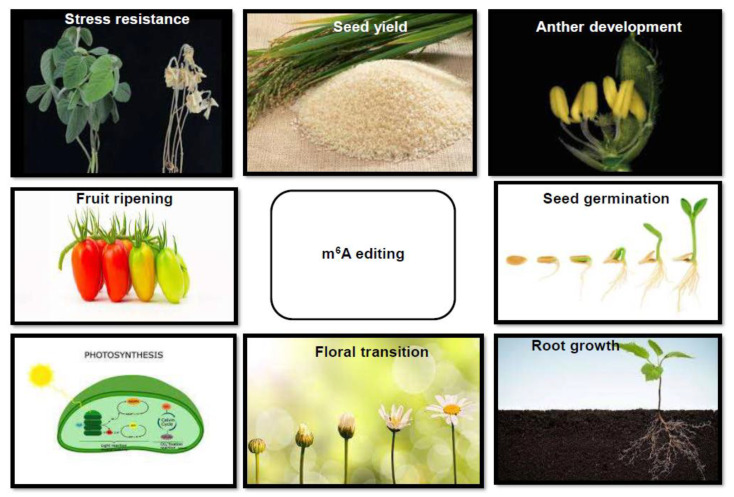
Improving useful traits through m^6^A editing; Various plant developmental processes such as seed development, anther development, root growth, floral transition, photosynthesis, fruit ripening, seed yield, and stress resistance can be improved through m^6^A editing.

**Table 1 plants-11-01033-t001:** Role of post-transcriptional modifications in plants’ growth and development.

Gene	Modification	Developmental Role	Plant Species	Reference
*TRM61/TRM6*	tRNA (m^1^A)	Embryogenesis	*Arabidopsis thaliana*	[53,65]
Complex				
*PhTRMT61A*	mRNA (m^1^A)	leaf development	*Petunia*	[66]
*AtTRM5*	tRNA	leaf and root development	*Arabidopsis thaliana*	[67]
	(m^1^G, m^1^I)	flowering time regulastion		
*TCTP1*	mRNA (m^5^C)	root growth	*Arabidopsis thaliana*	[68]
*FIP37*	mRNA (m^6^A)	embryo development	*Arabidopsis thaliana*	[69]
		trichome endoreduplication and shoot stem cell fate		[16,70]
*OsEMD2L*	mRNA (m^6^A)	anther development	*Oryza sativa*	[71]
*ECT2*	mRNA (m^6^A)	trichome branching,	*Arabidosis thaliana*	[33,72]
		trichome morphology		
*MTA*, *MTB,*	mRNA (m^6^A)	vascular formation in	*Arabidopsis thaliana*	[73]
*FIP37*, *VIR*,		roots, pattern formation		
*HAKAI*				
*MTA*, *MTB*	mRNA (m^6^A)	fruit ripening	*Fragaria ananassa*	[74]

**Table 2 plants-11-01033-t002:** Role of post-transcriptional modifications in stress responses.

Modification	Species	Stress	Regulatory Role	Reference
rRNA	*Arabidopsis thaliana*	Low	Confers cold stress tolerance	[64]
(m^2^G)		temperature	by resulting translation of chloroplast proteins including *RbcL*, *AtpB*, *PsbA*, *Ycf3*, and *PetC*	
mRNA	*Oryza*	high	Confers heat acclimation through regulating translation of transcripts involved in photosynthesis and detoxification such as *β- OsLCY*, *OsHO2*, *OsPAL1*, and *OsGLYI4*	[103]
(m^5^C)	*Sativa*	temperature		
tRNA	*Oryza*	salinity	Confers salts stress tolerance by	[59]
(Am)	*s* *ativa*		regulating the expression level of ABA-related (*SnRK2.1*, *ABA1*, *ABI5*, *AAO3*, and *RCAR1*) and salt-related (*HKT1*, *NHX1*, and *SOS1)* genes	
mRNA	*Sorghum*	salinity	Confers salts stress tolerance by regulating the mRNA stability of several stress-responsive transcripts including AVP1 and IAR4	[35]
(m^6^A)	*bicolor*			
mRNA	*Gossypium*	salinity	Confers salt stress tolerance by stabilizing the salt-responsive gene transcripts and regulating the levels of genes involved in zeatin biosynthesis, taurine and hypotaurine metabolism, and ribosome and proteasome processes	[104]
(m^6^A)	*hirsutum acc.*			
mRNA	*Populus*	drought	Confers drought resistance by affecting trichome and root development	[83]
(m^6^A)	*trichocarpa*			
mRNA	*Arabidopsis thaliana*	bacterial	Induced pattern-triggered immunity (PTI) and salicylic acid (SA)-mediated immune responses through an enhanced abundance of defense-related transcripts	[105]
(m^6^A)		attack		

## Data Availability

Not applicable.

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
