# Peer review of "Epitranscriptomics: An Additional Regulatory Layer in Plants’ Development and Stress Response"

_plants, 2022, doi:10.3390/plants11081033_

Round 1

Reviewer 1 Report

The paper entitled Epitranscriptomics: An Additional Regulatory Layer in Plants’ Development and Stress Response' brings good quality information on the role of Epitranscriptomics in plant development and stress biology. The authors have provided an in-depth study on the roles of epitranscriptomic modifications in modulating molecular, physiological, and stress responses in plants. However specific comments are furnished below which need to be addressed.

  1. Abstract needs to be improved as it is too general.It should highlight its significance, limitations and future perspectives. 
  2. Introduction part can be improved. I suggest authors to highlight why Epitranscriptomics is important than other regulatory mechanisms in plants. 
  3. I would suggest authors to mention any functional validation based on Epitranscriptomics in model or crop plants with case study.
  4. What are the limitations to fully harness the potential of Epitranscriptomics
  5. How Epitranscriptomics can be used for modern breeding and genetic engineering to improve crop yield in sustainable agriculture in the face of climatic challenges and food security threats.
  6. How Epitranscriptomics regulates signal transduction pathways during plant development and stress. Is there any known model which shows how it controls hormonal cascades, ROS, calcium etc. Authors can add a figure.
  7. Figures can be improved with more resolution and also improve figure legends.
  8. I would suggest authors to split the unnecessary lengthy sentences so that it will be more meaningful. The overall English and general punctuation of the manuscript could be improved. After carefully reading this manuscript I recommend for minor revision.

Author Response

The paper entitled Epitranscriptomics: An Additional Regulatory Layer in Plants’ Development and Stress Response' brings good quality information on the role of Epitranscriptomics in plant development and stress biology. The authors have provided an in-depth study on the roles of epitranscriptomic modifications in modulating molecular, physiological, and stress responses in plants. However specific comments are furnished below which need to be addressed.

Response: We would like to thank the reviewer for nice and detailed comments and suggestions for the manuscript. We believe that the comments have identified important areas which are required improvement. After completion of the suggested edits, the revised manuscript has benefitted from an improvement in the overall presentation and clarity. We have tried to revise the whole manuscript and removed mistakes. Below, you will find a point-by-point description of how each comment was addressed in the manuscript.

Comment 1: Abstract needs to be improved as it is too general. It should highlight its significance, limitations and future perspectives. 

Response 1: We improved the abstract by adding the significance of epitranscriptomics in crop improvement (Lines 14-16), and limitations (Lines 18-23).

Comment 2: Introduction part can be improved. I suggest authors to highlight why Epitranscriptomics is important than other regulatory mechanisms in plants.

Response 2: We improved the introduction part by adding information about how epitranscriptomic modifications and the epitranscriptomic machinery are regulated with plant development and in different tissues (Lines 46-53). 

Comment 3: I would suggest authors to mention any functional validation based on Epitranscriptomics in model or crop plants with case study.

Response 3: Individual research reports are added in different sections of development and stress response, for example, it is mentioned that transgenic expression of human-FTO increased yield in rice (Lines 477-482) and over-expression of  PtrMTA increased the m6A levels in poplar plant and enhanced trichome branching in roots and improve its survival and productivity (lines 482-485). CIMTB (an m6A methyltransferase) helps in the adaptation to drought stress in watermelon by regulating the ROS and multiple stress responsive pathways (Lines 292-294). In apple, the transgenic expression of MhYTP2 (an m6A reader protein) conferred the resistance to powdery mildews (lines 322-324). In addition, various case studies are listed in Table 2.

Comment 4: What are the limitations to fully harness the potential of Epitranscriptomics.

Response 4: We added the limitations about harnessing the epitranscriptomics in crop engineering in future perspectives (Lines 440-448, 450-452)

Comment 5: How Epitranscriptomics can be used for modern breeding and genetic engineering to improve crop yield in sustainable agriculture in the face of climatic challenges and food security threats.

Response 5: The potential of using epitranscriptomics for crop improvement through editing m6A or m5C modifications in the transcripts specific for development and stress response as written in Lines 458-461.

Comment 6: How Epitranscriptomics regulates signal transduction pathways during plant development and stress. Is there any known model which shows how it controls hormonal cascades, ROS, calcium etc. Authors can add a figure.

Response 6: Various processes of RNA metabolism (Figure 2) (Lines 382-385) and ABA response (line 172-174) are regulated by epitranscriptomic modifications but the signaling cascades or models of how these pathways are regulated by epitranscriptomic modifications have not been fully uncovered yet. There is still a need to understand the role of epitranscriptomics in regulating signaling pathways in plants through further analyses.

Comment 7: Figures can be improved with more resolution and also improve figure legends.

Response 7: We revised the figures and legends (Lined 223-226, 387-390, 431-434, 474-476)

Comment 8: I would suggest authors to split the unnecessary lengthy sentences so that it will be more meaningful. The overall English and general punctuation of the manuscript could be improved. After carefully reading this manuscript I recommend for minor revision.

Response 8: This manuscript got the professional edition before submission, and we have a certification document. Through this revision. we improved the overall English and punctuation and split the lengthy sentences. We hope that the present version of the manuscript is satisfactory. We again thank the reviewer for the nice comments, suggestions, and for giving valuable time to review our manuscript.

Reviewer 2 Report

The intent of this review is to highlight the significance of epitranscriptomics in the regulation of many cellular processes in plants, such as seed development, root development, and floral regulation. The review is thorough and well-presented.

Minor comments:

Line 190-191: The sentence is incomplete. See below. 

“The deletion of vir-1 showed aberrant root cap formation and defective protoxylem development, indicating that [67].”

Line 363: “form6A”

Please add a space between “form” and “6A”

Line 373” “theyregulate”

Please add a space between “they” and “regulate”

Line 448: “resolutionwith”

Please add a space between “resolution” and “with”

Author Response

The intent of this review is to highlight the significance of epitranscriptomics in the regulation of many cellular processes in plants, such as seed development, root development, and floral regulation. The review is thorough and well-presented.

Response: We are thankful to the reviewer for positive comments on our manuscript. We have taken the comments on board to improve and clarify the manuscript. Below is our responses to the comments raised by you.

Minor Comments:

Comment 1: Line 190-191: The sentence is incomplete. See below. 

“The deletion of vir-1 showed aberrant root cap formation and defective protoxylem development, indicating that [67].”

Response 1: We revised and completed the sentence (line 208)

Comment 2: Line 363: “form6A”

Please add a space between “form” and “6A”

Response 2: We revised and added the space between for and m6A (line 395)

Comment 3: Line 373” “theyregulate”

Please add a space between “they” and “regulate” 

Response 3: We revised and added the space between they and regulate (line 406)

Comment 4: Line 448: “resolutionwith”

Please add a space between “resolution” and “with”

Response 4: We revised and added the space between resolution and with (line 491)

We have checked the whole manuscript again and made all possible corrections. We hope that this version of the manuscript is more balanced and reader-friendly. Thanks for the constructive suggestions and for reading our manuscript carefully.

Reviewer 3 Report

This review article summarizes recent studies that link modification-induced transcript regulation to important biological outcomes such as development and stress response in plants and design some important perspective in crop breeding through genome editing.

The manuscript attended its purpose and presented very interesting information about the subject

Author Response

This review article summarizes recent studies that link modification-induced transcript regulation to important biological outcomes such as development and stress response in plants and design some important perspective in crop breeding through genome editing.

The manuscript attended its purpose and presented very interesting information about the subject

Response: Thank you for reading our manuscript carefully and giving positive feedback and comments. We appreciate the time and effort that you have dedicated to providing your valuable feedback on our manuscript.

Reviewer 4 Report

Comments on review „Epitranscriptomics: An Additional Regulatory Layer in Plants’ Development and Stress Response“ by Shoaib et al., submitted to MDPI Plants

The authors’ review aims to summarize the state of research on RNA modifications with a focus on m5C cytidine methylation and, yet more importantly, on m6A adenosine methylation as prevalent mRNA modifications. The manuscript is well written. In their selection of 111 references, the authors seem to emphasize literature reports that make connections to plant developmental processes, possibly somewhat at the expense of others that give more insights on the biochemical machinery behind epitranscriptome “writing” and “erasing” enzymes, but this is certainly the authors’ choice and freedom. They may, however, ultimately wish to consider toning down statements making direct links from the plant epitranscriptome to securing food security and fighting world hunger, which to me seem to be a little far-fetched and going overboard.

Other than the above general remark I have the following detailed comments:

  1. The terms “RNA modification” and “RNA editing” are not sharply defined, but most researchers would likely agree that the former addresses creation of non-standard nucleotides (like the ones under consideration  here) whereas the latter processes deal with conversion, insertion and deletion of the for standard ribonucleotides (see e.g. Knoop, 2011). A semantically overlapping case is the deamination of adenosine to inosine (abundant in animals), which is considered RNA editing since inosine behaves very much like guanosine. This, in fact, is the only reference to the term "RNA editing" in the manuscript (line 70). For clarity, the authors should at least mention that the abundant plant organelle C-to-U and U-to-C RNA editing exists (see e.g. Small et al, 2020), but is not subject of this review although certainly a “post-transcriptional process that may be considered “epitranscriptomic”, too.

  2. Along the same lines, clear statements on which processes are identified in which genetic compartment of the plant cell (nucleo-cytosolic, chloroplast or mitochondrial) would be very helpful to the reader new to the subject. Chloroplast and mitochondrial examples are mentioned somewhat "in passing" in the paper.

  3. Likewise, in the interest of unequivocal clarity, and since the authors also refer to genomic engineering it should always be made clear whether actually RNA/transcripts or, occasionally, DNA/genes are referred to. A typical example would be “methylation profile in genes” in line 185 or “differentially methylated genes” in line 296.

  4. Throughout the paper references should be placed directly at the end of sentences where important statements on observations are made. Typical examples (by no means exhaustive !) are sentences like the ones starting “A recent study…” on line 136 and on line 164, “Another research…” on line 144, “A recent report…” on line 257 or “Recent research showed…” on line 307.

  5. Figure 1 should include the respective references behind the genes mentioned.

  6. Related to my introductory comments: Are really so few details on the biochemical mechanisms known. In other words: Are there no interesting reports on what the immediate molecular effects of decreased or increased epitranscriptomic modifications are and what exactly the links to the observed phenotypes are? At least a few more references in that direction could make the paper a yet more interesting reference to be cited by interested colleagues. Possibly, a figure on the immediate effects of m6A (and m5C) could even be more valuable than the current colorful but not highly informative “advertising” figure 3.

  7. Related to the above, I cannot quite understand the direction of arrows in figure 2. Isn’t it that rather the epitranscriptomic markers have an effect on phenotypes than the other way around as the current direction of arrows seems to suggest.

Again, however, aside from a few remaining typos (e.g. reHulating in line 323 or “accliAmation” in line 326) the paper reads quite well and it is certainly the authors' choice to focus on selected aspects of plant epitranscriptomics.

References:

Knoop V (2011) When you can’t trust the DNA: RNA editing changes transcript sequences. Cell Mol Life Sci 68: 567–586

Small ID, Schallenberg-Rüdinger M, Takenaka M, Mireau H & Ostersetzer-Biran O (2020) Plant organellar RNA editing: what 30 years of research has revealed. Plant J 101: 1040–1056

Author Response

The authors’ review aims to summarize the state of research on RNA modifications with a focus on m5C cytidine methylation and, yet more importantly, on m6A adenosine methylation as prevalent mRNA modifications. The manuscript is well written. In their selection of 111 references, the authors seem to emphasize literature reports that make connections to plant developmental processes, possibly somewhat at the expense of others that give more insights on the biochemical machinery behind epitranscriptome “writing” and “erasing” enzymes, but this is certainly the authors’ choice and freedom. They may, however, ultimately wish to consider toning down statements making direct links from the plant epitranscriptome to securing food security and fighting world hunger, which to me seem to be a little far-fetched and going overboard.

Response: We would like to thank the reviewer for positive and insightful comments on the manuscript. We believe this input has been invaluable to make our manuscript more balanced. We have taken the comments on board to improve and clarify the manuscript. Below is our response to the comments raised in the review.

Comment 1: The terms “RNA modification” and “RNA editing” are not sharply defined, but most researchers would likely agree that the former addresses creation of non-standard nucleotides (like the ones under consideration here) whereas the latter processes deal with conversion, insertion and deletion of the for standard ribonucleotides (see e.g. Knoop, 2011). A semantically overlapping case is the deamination of adenosine to inosine (abundant in animals), which is considered RNA editing since inosine behaves very much like guanosine. This, in fact, is the only reference to the term "RNA editing" in the manuscript (line 70). For clarity, the authors should at least mention that the abundant plant organelle C-to-U and U-to-C RNA editing exists (see e.g. Small et al, 2020), but is not subject of this review although certainly a “post-transcriptional process that may be considered “epitranscriptomic”, too.

Response 1: We added information about C-to-U conversion and its effects on RNA structure, sequence and stability in the mitochondria and chloroplast (lines 73-76) and revised the (line 91) and added definition of epitranscriptomics (line 30).

Comment 2: Along the same lines, clear statements on which processes are identified in which genetic compartment of the plant cell (nucleo-cytosolic, chloroplast or mitochondrial) would be very helpful to the reader new to the subject. Chloroplast and mitochondrial examples are mentioned somewhat "in passing" in the paper.

Response 2: Information about the processes regulated by epitranscriptomic modifications in nucleus and cytoplasm are added in lines 65-66 and in chloroplast and mitochondria in lines 69-73.

Comment 3: Likewise, in the interest of unequivocal clarity, and since the authors also refer to genomic engineering it should always be made clear whether actually RNA/transcripts or, occasionally, DNA/genes are referred to. A typical example would be “methylation profile in genes” in line 185 or “differentially methylated genes” in line 296.

Response 3: We revised the lines 202 and 318.

Comment 4: Throughout the paper references should be placed directly at the end of sentences where important statements on observations are made. Typical examples (by no means exhaustive !) are sentences like the ones starting “A recent study…” on line 136 and on line 164, “Another research…” on line 144, “A recent report…” on line 257 or “Recent research showed…” on line 307.

Response 4: These references are added at the end of the mentioned details (lines 154, 162, 186, 282, 328).

Comment 5: Figure 1 should include the respective references behind the genes mentioned.

Response 5: Reference for the genes indicated in Figure 1 are added in Lines 219-221.

Comment 6: Related to my introductory comments: Are really so few details on the biochemical mechanisms known. In other words: Are there no interesting reports on what the immediate molecular effects of decreased or increased epitranscriptomic modifications are and what exactly the links to the observed phenotypes are? At least a few more references in that direction could make the paper a yet more interesting reference to be cited by interested colleagues. Possibly, a figure on the immediate effects of m6A (and m5C) could even be more valuable than the current colorful but not highly informative “advertising” figure 3.

Response 6: Figure 2 is added to address the regulation of the immediate molecular effects occurring after the transcript modification that depends on the reader protein recognizing the m6A modification (lines 382-390).

Comment 7: Related to the above, I cannot quite understand the direction of arrows in figure 2. Isn’t it that rather the epitranscriptomic markers have an effect on phenotypes than the other way around as the current direction of arrows seems to suggest.

Response 7: The direction of arrows is corrected in the Figure 3 by showing the molecular and physiological processes regulated by epitranscriptomic modifications.

Comment 8: Again, however, aside from a few remaining typos (e.g. reHulating in line 323 or “accliAmation” in line 326) the paper reads quite well and it is certainly the authors' choice to focus on selected aspects of plant epitranscriptomics.

Response 8: Lines 344 and  347 are revised, and typo errors are fixed.